# Aspects of Efficiency Enhancement in Reflectarrays with Analytical Investigation and Accurate Measurement

**M. Hashim Dahri [1], M. Haizal Jamaluddin [2],\* , Fauziahanim C. Seman [1],\* , M. Inam Abbasi [3] , N. Fazreen Sallehuddin [2], Adel Y. I. Ashyap [1] and M. Ramlee Kamarudin [1]**

[1] Faculty of Electrical and Electronic Engineering, Universiti Tun Hussein Onn Malaysia (UTHM), Batu Pahat 86400, Johor, Malaysia; muhammadhashimdahri@yahoo.com (M.H.D.); ashyap2007@gmail.com (A.Y.I.A.); mramlee@uthm.edu.my (M.R.K.)

[2] Wireless Communication Centre of Universiti Teknologi Malaysia (UTM), Johor Bahru 81310, Malaysia; fazreen83@yahoo.com

[3] Centre for Telecommunication Research & Innovation (CETRI), Faculty of Electrical and Electronic Engineering Technology (FTKEE), Universiti Teknikal Malaysia Melaka (UTeM), Melaka 76100, Malaysia; inamabbasi@utem.edu.my

\* Correspondence: haizal@fke.utm.my (M.H.J.); fauziahs@uthm.edu.my (F.C.S.)

**Abstract:** This paper presents a thorough review of the techniques involved in the enhancement of the efficiency performance of the reflectarray antenna. The effect of the selection of a suitable patch element or a proper feeding mechanism on efficiency improvement is studied in detail. Reflectarray loss quantification is examined in relation to the design techniques involved in the efficiency improvement. A low loss patch element with a wide reflection phase range and a properly illuminated reflectarray aperture are supposed to offer high efficiency performance. Additionally, the placement, the orientation and the position of a patch element on the reflectarray surface can also affect its efficiency performance. Mathematical equations were developed to estimate the efficiencies of circular and square aperture reflectarray antennas by considering their feed footprints. Moreover, a step by step practical method of predicting and measuring the total efficiency of a reflectarray antenna is presented. The two selected apertures of the reflectarray consisting of the square patch element configuration are fabricated and measured at a frequency of 26 GHz. Their measured efficiencies have been estimated using the derived equations, and the results were compared and validated using the efficiencies obtained by the conventional gain-directivity relation.

**Keywords:** aperture efficiency; circular aperture; feed footprint; reflectarrays; square aperture; total efficiency

## 1. Introduction

The array of resonating elements used to reflect the incoming signals from a suitably placed feed, defines the basic architecture of a reflectarray antenna [1,2]. It evolved from the combination of the concepts of a parabolic reflector and a phased array antenna. It holds the advantage of a flat and light structure over the parabolic reflector, whereas its simple design overcomes the flaws of the lossy phased arrays [3]. Reflectarray can attain high gain performance like a parabolic reflector and it can scan its beam over wide angles just like a phased array antenna [4]. All this can be achieved without adding any extra power dividing or phase shifting mechanism. A reflectarray can be of many types, including an array of reflecting waveguides [5] or a conventional reflecting surface of the microstrip patch array [1]. A full metallic reflectarray with variable size grooves [6] and a full dielectric reflectarray with implanted air holes [7] have also been introduced to overcome the effects of

extra material losses [8]. The basic architecture of an offset feed square patch microstrip reflectarray is shown in Figure 1. The printed microstrip patches on a dielectric substrate are used to reflect the incident signals of a distant feed. The reference element on the reflectarray indicates a point with zero phase difference to reflect the incident signals. This phase difference progresses as the distance between the patches and the feed increases beyond the reference element. In order to maintain this phase difference, various techniques can be found in the literature. It can be done using patches with variable sizes [1,9], by using a variable rotational angle [10,11] or by adding an extra phase tuning slot or stub into them [12,13]. The main reason to maintain this phase difference is to reflect all the incoming signals in the same direction with the same phase [14]. In order to attain this progressive phase, a full wave analysis of the reflectarray is required by considering a single element as a unit cell [3]. The boundaries of the unit cell, as shown in Figure 1, determine whether the effects of its surrounding elements are considered or not. The infinite boundary conditions of the unit cell with electric and magnetic walls ensure the accumulation of the mutual coupling effects of the surrounding elements [15]. The beamwidth of the reflectarray element is also an important factor, which describes its ability to reflect incident signals. A wider element beamwidth is required to properly reflect the signals from the corner of the reflectarray.

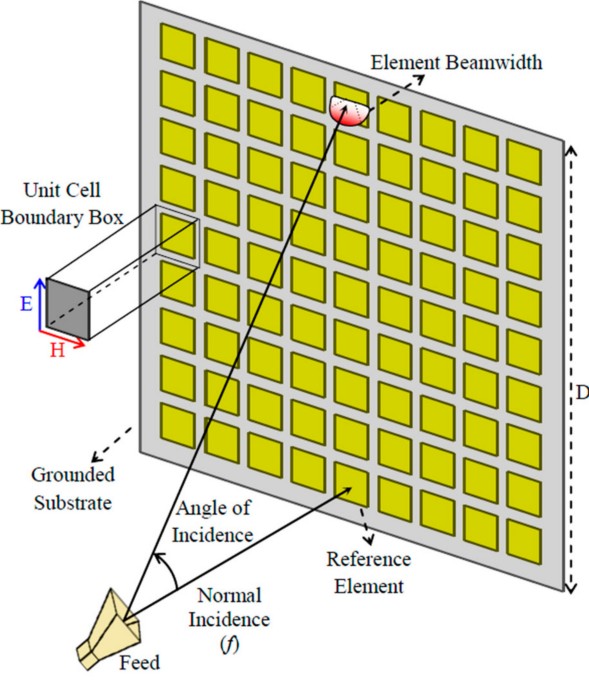

**Figure 1.** The basic architecture of a microstrip reflectarray with an offset feed.

In order to achieve the best possible performance, a reflectarray antenna should be designed carefully. However, even a perfectly designed reflectarray would still lack in its bandwidth and gain performance as compared to a parabolic reflector [3,16]. The theoretically infinite bandwidth of a parabolic reflector is due to its non-resonant structure, while its higher gain performance is because of its full metallic design. One of the main reasons a reflectarray is preferred over a parabolic reflector is its ability to acquire electronic beamsteering with a comparatively small antenna profile [4]. This could also be done with a phased array antenna, but its design complexity is a major issue [17], which limits its progression towards millimeter waves. The extra electronic components attached in the phased array also escalate its loss performance and reduce its efficiency. The gain of a reflectarray antenna can also be enhanced using a full metallic design or a curvy surface just like a parabolic reflector [6,18]. However, it can increase the design complexity and destroys the purpose of acquiring the electronic beamsteering. Figure 2 compares the reflectivity phenomenon of a microstrip reflectarray and a

parabolic reflector. Figure 2a shows that the reflection of the incident electric field ($E_i$) from the surface of the reflectarray occurs in two different ways—the first one due to its resonant behavior ($E_{rr}$) and the second one due to its non-resonant behavior ($E_{rn}$). The resonance depends on the material properties, substrate thickness and patch dimensions of the reflectarray, whereas the ground plane is responsible for its non-resonant behavior. This unwanted non-resonant behavior drives the reflected electric field away from the desired direction of the reflection. This effect of non-resonance is very small in reflectarrays, but for an optimized performance it cannot be neglected. On the other hand, a parabolic reflector totally works on the principle of non-resonant behavior and reflects the incident electric field in the desired direction, as depicted in Figure 2b. This is the main reason behind the higher gain performance of the parabolic reflector as compared to the reflectarray antenna.

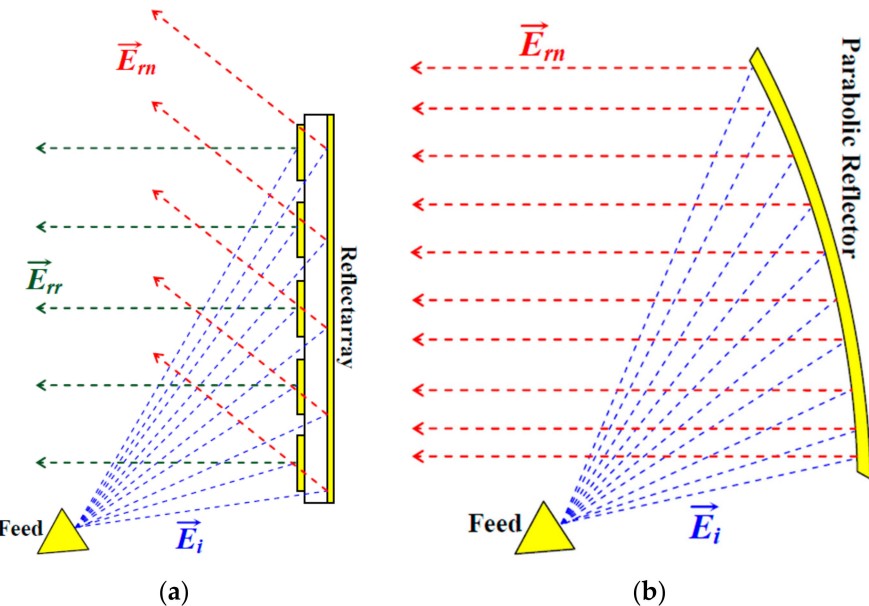

**Figure 2.** Reflection of the incident signals from the surface of the: (**a**) reflectarray and (**b**) parabolic reflector.

The lower gain performance contributes to reducing the efficiency of the reflectarray antenna. It implies one should consider 50% or more as a high efficiency for reflectarray antenna [19,20]; alternatively, this scenario is completely different for a parabolic reflector. Figure 3 relates the efficiency scenario of the reflectarray antenna with the parabolic reflector. It has been shown here that the quality of acquiring high efficiency in reflectarrays lags behind the efficiency performance of the parabolic reflector. However, it does not restrain reflectarray from achieving high efficiency performance [21,22]. If the reflectarray is optimized and designed properly, it can achieve an efficiency of around 60% [23].

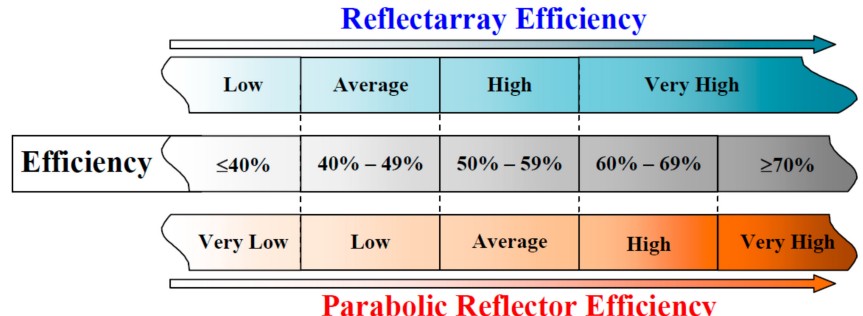

**Figure 3.** Efficiency comparison between reflectarray and parabolic reflector.

Reflectarray antenna efficiency is its most complex parameter to optimize properly. In most cases it is just calculated using the simple gain-efficiency equation [24]. However, a complete quantification of the reflectarray loss performance is required to clearly understand the real picture of reflectarray efficiency. The efficiency of the reflectarray antenna also depends on its feeding mechanism, which is related to its aperture area under its feed footprint. Moreover, the conventional way to calculate reflectarray efficiency considers only the circular shape apertures [3,17]. The whole scenario of the efficiency and loss mechanism would be different for a reflectarray designed with a different aperture shape. The analytical investigation of reflectarray aperture efficiency is also provided in [25], but it does not consider the total efficiency performance with supported simulations and practical measurements. The originality of this research lies in the detailed and thorough explanation of the most complex parameter of the reflectarray antenna performance—that is, its efficiency. The concepts discussed in this work will help the researchers to significantly save their design time by easily estimating the total efficiency of any type of reflectarray antenna even before its simulation and measurements are carried out. It will also help them to build a clear idea for different possibilities of efficiency enhancement in their planned design. Therefore, in this article, a detailed analysis involving all the aspects of the reflectarray antenna efficiency performance is provided. CST MWS simulations and the analytical solution of the mathematical equations were used to predict the required results. Moreover, some points regarding reflectarray efficiency, which are confusing and not clearly discussed in textbooks, are also explained in this article. These points include the effect of the feed footprint on reflectarray efficiency, the formulation of the reflectarray subtend angle and understanding the relationship between feed pattern function ($q$) and reflectarray aperture efficiency. Section 2 thoroughly discusses the sources of all the possible losses in reflectarray antenna and their effect over the efficiency performance. The background studies explaining some common efficiency enhancement techniques in reflectarrays are provided in Section 3. The factors other than the losses that affect the reflectarray antenna efficiency are analyzed in Section 4. In Section 5, a new mathematical formulation has been derived for the efficiency prediction of the square aperture reflectarray and its results are compared with the circular aperture reflectarray. Finally, in Section 6, the practical method of accurately predicting and measuring the efficiency performance of a reflectarray antenna is presented by performing the radiation pattern measurements of square and circular aperture microstrip reflectarrays operating at 26 GHz frequency.

## 2. Loss Quantification of Reflectarray Antenna

The proper analysis of all the losses associated with the reflectarray antenna performance primarily requires the understanding of the gain and maximum directivity relation. Maximum directivity expresses the peak reflected power density of the reflectarray antenna without accumulating its loss performance. Alternatively, all the losses in the reflectarray antenna are considered to predict its gain performance. Therefore, the logarithmic difference between the gain and maximum directivity of the reflectarray antenna accounts for the actual amount of losses involved in its operation. Maximum directivity ($D$) [25,26] and gain ($G$) are distinguished using the following common relations [24].

$$G = \frac{4\pi A_{eff}}{\lambda^2} = \frac{4\pi \eta A}{\lambda^2} \tag{1}$$

$$D = \frac{4\pi A}{\lambda^2} \tag{2}$$

$$\eta = \frac{G}{D}(Magnitude) = G - D \ (dB) \tag{3}$$

where $A$ and $A_{eff}$ are the aperture area and the effective aperture area of the reflectarray respectively, $\lambda$ is the free space wavelength and $\eta$ is the efficiency.

The losses of the reflectarray antenna can be divided into three different categories, feed loss, patch element loss and aperture loss. The losses associated with the feed depend on its type,

material properties and radiation characteristics. If the feed is the selected properly, then these losses are often considered harmless and possess a very small effect on the reflectarray performance. The patch element is a major contributor to losses in the reflectarray antenna. The conductor and dielectric losses are the material losses linked with the reflectarray patch element [27]. A delay line, if attached to the patch element for phase tuning purpose, can also produce an extra amount of unwanted loss. The dissimilarity or asymmetry in the patch design can introduce cross polarization losses to the reflectarray antenna. The reflectivity of the patch element can also be deteriorated if a defected ground plane is used for any expected improvement. The amount of loss associated with the patch element shows the importance of its design sensitivity. Once a reflectarray antenna is designed, its feed and patch losses will remain constant for throughout its operation.

The patch element losses in reflectarray can be reduced using sub-wavelength elements on its surface [28]. Commonly, the distance between the patch elements on the reflectarray surface is selected as half of the wavelength ($0.5\lambda$) [9]. If this distance is reduced between the patches then a large number of patches can be occupied in the same physical aperture area, which was initially selected for the half wavelength spaced elements. In this way a significant reduction in the patch element losses can be reported, because most of the incident signals would be reflected back by the patch and only a small number of them would interact with the substrate and dissipate within it [29]. It will also decrease the capacitive dissipation of incident energy within the ground based substrate due to the reduction in its surface area [30]. The sub-wavelength elements with an element spacing of $0.4\lambda$, $0.3\lambda$ or even less can be used for the described purpose [31,32]. However, this element spacing should be greater than the maximum dimension of the metallic patch element; otherwise, the metallic patches would overlap each other on the reflectarray surface. A significant amount of element loss reduction is reported in [28] when the sub-wavelength elements are used in place of the conventionally spaced elements in the reflectarray. This loss reduction is also accompanied by the reduction in the capability of attaining a higher progressive phase range by the element. This reduction in the progressive phase range of the element is called phase error [33,34], which happens because of the reduction in the available space required by the patch element to vary its overall size [1]. This problem can be avoided by selecting an element with high design flexibility, such as a rectangular ring element, as proposed in [32]. This rectangular ring element can be selected to reduce the overall size of the reflectarray by imposing a sub-wavelength technique for loss reduction without a significant reduction in the gain performance. However, this reduction in the loss is accompanied by a slight reduction in the bandwidth performance of the reflectarray. The elements like rings and loops are preferred as a useful technique to drastically improve the efficiency performance of the reflectarray antenna using a sub-wavelength element approach with a reduction in its electrical aperture area [35]. It should also be noted that the patches such as the square ring and circular ring would practically offer higher reflection losses as compared to a square patch or circular patch elements [36]. The sub-wavelength technique physically reduces the distance between the adjacent metallic patch elements that can induce an additional amount of inductive loading [29]. This additional inductive loading could produce a mutual coupling issue, which can be tackled by designing the element using the Floquet technique that takes into account the arising effect of mutual coupling [3].

The other major contributor to the reflectarray losses is the aperture loss, which includes spillover and illumination effects [21]. The spillover and illumination losses are complementary to one another and depend on the feed position and distance in front of the reflectarray, as highlighted in Figure 4. It can be observed from Figure 4a that if the feed is located very far from the reflectarray, a portion of incident signals would be wasted by passing beyond the edges of the reflectarray. In this case, the wasted energy is the spillover loss of the reflectarray antenna. Alternatively, if the feed is situated very close to the reflectarray, as depicted in Figure 4b, then a portion of the reflectarray would remain unilluminated, causing illumination loss. Aperture loss is variable in amount and can be optimized by adjusting a suitable feed distance, even after the designing of the reflectarray. Therefore, the proper

optimization of the aperture loss is very important in the reflectarray antenna design. The total efficiency of a reflectarray antenna can be predicted if all its losses are quantified accurately.

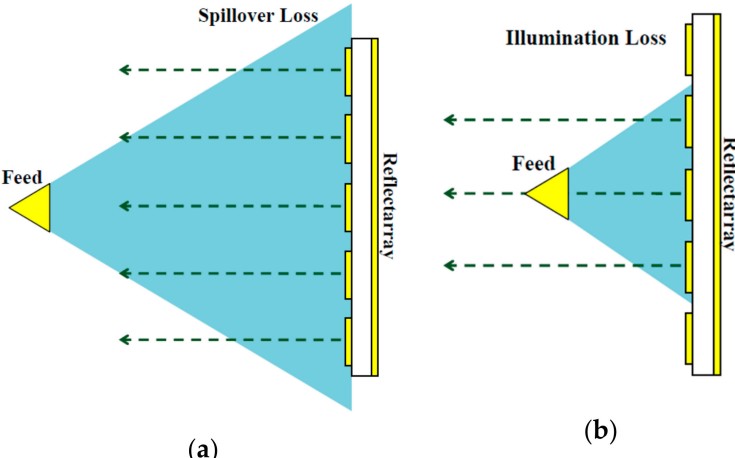

**Figure 4.** Sources of the aperture loss in the reflectarray antenna: (**a**) spillover loss; (**b**) illumination loss.

## 3. Background of the Common Techniques of Efficiency Improvement in Reflectarrays

The enhancement in the efficiency of the reflectarray antenna is related to the improvement in its gain performance. A simple way to increase the gain of an antenna is to enlarge its physical aperture size [37]. However, a large physical aperture of reflectarray can induce an additional amount of illumination or spillover loss if its feed is not positioned properly [17], which is not good for efficient operation. This shows that maximum efficiency can be achieved from a reflectarray antenna if its losses are minimized, even with the same gain performance [24]. Additionally, a low side lobe level with a negligible amount of cross polarization is also essential for highly efficient operation [17]. The reduction in the side lobe level is linked with a proper distribution of the progressive phase range over the reflectarray surface with minimum aperture loss. On the other hand, the effect of cross polarization can be minimized with a suitably shaped reflectarray patch element that does not offer any asymmetry in its design [38]. Consequently, it can be said that the efficiency of a reflectarray antenna can be improved by minimizing its element loss and aperture loss. Therefore, the enhancement in the efficiency of the reflectarray antenna can be analyzed based on a suitable unit cell element or by a proper full reflectarray structure.

### 3.1. Elements with Low Loss and Wide Reflection Phase Range

A unit cell element with negligible loss is the best choice to select for efficient performance of the reflectarray antenna. However, there is a tradeoff between the loss and phase range performance of the reflectarray unit cell element [39]. Generally, a low loss element would suffer with a narrow phase range performance. A wide reflection phase range is essential to design a reflectarray antenna with high precision [40]. Patch elements such as all types of rings are commonly known for their smooth and wide reflection phase rangs, but have high-loss performances [39]. Alternatively, low loss patch elements, such as square, rectangular and circular ones, offer narrow phase ranges with a lot of phase error [34,41]. Therefore, the strategic design of a unit cell patch element that acquires a wide reflection phase range with a low loss performance is required for efficiency enhancement. There are many proposed element types available in the literature for said purpose; some of them are discussed here to portray a basic idea of their contribution to efficiency improvement.

Table 1 contains the list of some patch elements from the literature that have been proposed for the high efficiency reflectarray operation. A reflectarray antenna is said to offer a high efficiency operation if its aperture efficiency exceeds the 50% benchmark. Table 1 shows that a reflectarray antenna with high efficiency can be designed with a patch element of low loss and wide reflection phase range.

All the elements listed in Table 1 attain a reflection phase range of 360° or higher with a reflection loss of less than −1 dB (wherever mentioned) that are utilized to attain a high efficiency performance (≥50%). A hexagonal element with cross slots, as reported in [42], is proposed to suppress the loss with a large reflecting surface and to attain a wide reflection phase range with the help of cross slots. The efficiency of the reflectarray antenna can also be enhanced with a circular polarization operation. The elements such as bow-tie [22,43] and I-shaped [20] can be used for this purpose. The multi-resonance technique for phase enhancement, as used in dual rings [44], three parallel dipoles [45], two Rings with patches [46] and three rectangular rings [23], is also essential for low loss operation due to the combination of more than one element in a single unit cell. A dipole element can be a good candidate for low loss operation, and a split ring is normally used to achieve a smooth phase span; both of these are combined in [47] to enhance the efficiency of the reflectarray antenna. A combination of elements for multi-resonance purposes can raise the issue of mutual coupling and degrade the reflectarray performance. A fractal element [48] can eliminate the issue of mutual coupling, as it is a single element used for wideband and low loss operation. The loss performance of a ting element can be reduced if it is used with multiple phase delay lines, as in [49], where the phase delay lines are also responsible for the attainment of the wide phase range. The main issue related to the elements of low loss and wide reflection phase range is the high design complexity that can restrain them from being used at millimeter wave frequencies.

**Table 1.** Reflectarray unit cell patch elements for highly efficient operation.

| Element Type | Design | Loss (dB) | Phase Range (°) | Aperture Size ($\lambda^2$) | Efficiency (%) |
|---|---|---|---|---|---|
| Hexagonal [42] | ⬓ | N.A | 360 | 69.4 | 60 |
| Bow-tie [22,43] | ✤ | −0.4 | 360 | 39 | 57 |
| I-Shaped [20] | (⬙) | N.A | 360 | 75.7 | 50 |
| Dual Rings [44] | ◎ | N.A | 360 | 250 | 52 |
| Parallel Dipoles [45] | ⦀ | −0.2 | 360 | 180 | 65 |
| Two Rings and Patch [46] | ▣ | N.A | 360 | 130 | 64 |
| Three Rings [23] | ▥ | N.A | 500 | 163 | 66 |
| Split Ring [47] | ⬚ | −0.3 | 360 | 187 | 55 |
| Fractal [48] | ▢ | −0.45 | 700 | 54.5 | 66 |
| Ring with Phase Delay Lines [49] | ◉ | −0.05 | 460 | 78 | 57.3 |

N.A = not available.

Apart from the mentioned element types, some other elements' techniques such as the insertion of a ground ring slot [50] and an independent amplitude control circuit [51] can also be used for low loss operation. The modification in the surface current and the generation of leaky waves due to the ground slot is essential to increase the gain performance of the reflectarray antenna. The amplitude control circuit is used as an impedance transform unit that is connected to a patch element to independently tune its loss and bandwidth performance without the involvement of the resonant frequency. This could be an essential tool for efficiency enhancement, but it requires a lot of design effort with an additional design cost.

### 3.2. Sub-Wavelength Elements for Loss Reduction

Another useful method to reduce the amount of loss in the reflectarray antenna is by reducing the inter-element distance between its elements. Elements that are involved in this technique are called sub-wavelength elements, as their sizes are further reduced from a conventional spacing of half of the wavelength [52]. The reason for loss reduction due to sub-wavelength technique was already explained in Section 2. A reflectarray antenna with sub-wavelength elements can be used for size reduction [53], bandwidth enhancement [52] and efficiency enhancement [54,55]. In this section, the emphasis has been given to the improvement of efficiency with the help of sub-wavelength elements in the reflectarray antenna.

It can be mentioned thanks to a notable work published in [42] that the loss of the reflectarray element is from −1 dB to −0.45 dB when the inter-element spacing is reduced from 0.5λ to 0.3λ. This loss reduction can easily contribute to enhancing efficiency performance. However, the reflection phase sensitivity is increased using the sub-wavelength technique, as it reduces the overall size of the element. The reduction in phase sensitivity can increase the side lobe level and reduce the gain performance of the reflectarray antenna. This issue can be tackled by multi-resonance elements. A single meander line [53], a double square meander line ring [55] and a complementary square ring [56] are the types of elements that offer low reflection phase sensitivity with sub-wavelength spacing for reflectarray development. Among the mentioned elements, the double square meander line ring element reflectarray [55] offers a highly efficient performance of 56.5% with 0.2λ of element spacing. The reduction in the spacing of the elements on the surface of reflectarray can cause the elements to experience the effect of mutual coupling [57]. A novel technique for mutual coupling reduction is proposed in [58], where an H shaped isolation wall is used between the elements to reduce the mutual coupling effect. This technique of mutual coupling reduction is very useful at low frequencies. However, at high frequencies of millimeter wave it can increase the design sensitivity and can cause fabrication errors.

Figure 5 depicts some examples of sub-wavelength-based reflectarrays. It can be observed from Figure 5a that a higher number of elements can be occupied in the same physical area with sub-wavelength technique [54]. An increment in the number of elements can reduce the element loss and increase the gain of the reflectarray antenna. This work demonstrates a significant improvement in the efficiency of the reflectarray antenna from 50.8% to 56.6% when 0.5λ spacing elements are replaced with 0.3λ spacing elements. The sub-wavelength technique is also useful to support highly efficient performance for circular polarization operation [59]. Figure 5b shows the circular polarization reflectarray elements with angular rotation that acquires an efficiency performance of 51.9%. Embedding sub-wavelength technique in reflectarray is an easy way to improve its efficiency. However, it requires high precision in fabrication at high frequencies when the number of reflectarray elements is also very large.

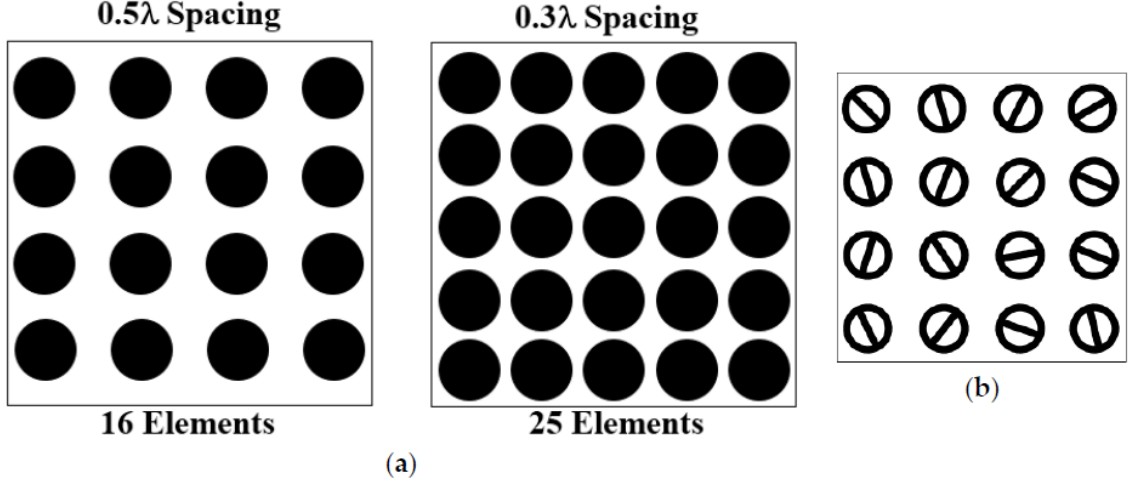

**Figure 5.** Sub-wavelength spacing in reflectarrays: (**a**) increment in number of elements due to sub-wavelength spacing; (**b**) sub-wavelength elements with angular rotation for circular polarization.

### 3.3. Strategic Feeding Mechanism for Aperture Loss Reduction

A perfectly designed feeding mechanism for a reflectarray antenna is considered as a backbone in its performance improvement with minimal aperture losses [60]. The main purpose is to offer a minimum aperture loss performance with possibly minimum cross polarization. The proper focusing of the incident signals on the reflectarray surface is very important to achieve high-efficiency operation

with minimum phase delays in the reflected signal [61]. In this case, a secondary sub-reflector with the main reflectarray is normally used in the literature for gain and efficiency improvement [62]. The basic architecture of a reflectarray antenna with a sub-reflector is shown in Figure 6a [63], which is used to reduce the feed based losses from the proposed structure. The feed, its sub-reflector and their main reflector should be placed within the near filed distance from each other to get the maximum advantage [62]. In order to get a maximum efficiency performance, both the reflectarrays should be designed carefully to eliminate any extra effect of cross polarization or additional aperture loss.

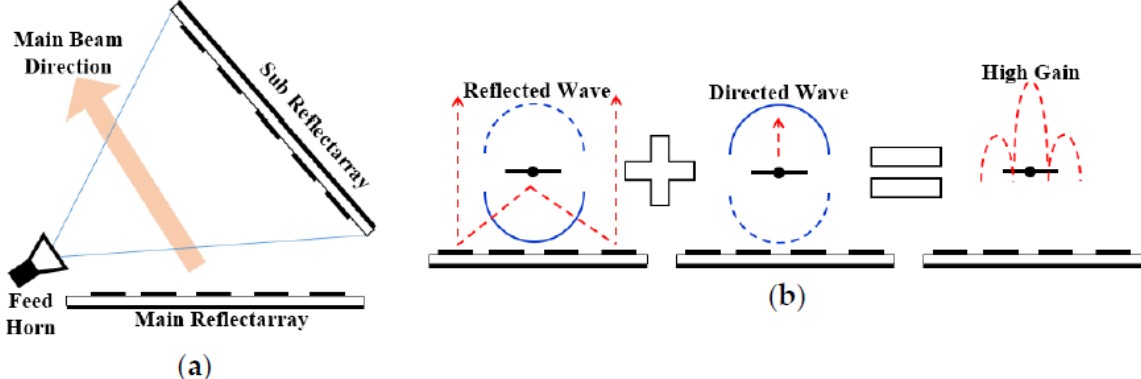

**Figure 6.** (**a**) Reflectarray with a sub-reflector; (**b**) the combination of radiated and reflected signals for efficiency improvement in reflectarrays.

The aperture loss of a conventional reflectarray antenna without the sub-reflector can also be optimized by varying its feed distance. The distance of a prime or an offset feed of a reflectarray antenna can be manually varied in order to observe its effects on the gain and efficiency enhancement [33,48]. Consequently, a suitable feed distance can then be set that acquires a maximal gain performance for the reflectarray antenna. This type of feed variation or using a sub-reflector can drastically increase the overall profile of the reflectarray antenna. The overall profile or size of a reflectarray antenna can be reduced by converting it into a folded reflectarray antenna [16]. However, in order to do so, one should have to compromise on the highly efficient performance of the reflectarray antenna. Gain or efficiency of a reflectarray antenna can also be increased if the radiation effects of the feed are combined with the reflected signals from the reflectarray [64]. Similar work is reported in [64] and is shown in Figure 6b, where a dipole antenna is used as a feed of a reflectarray to combine the radiated and reflected signals together for gain and efficiency improvement. This technique is also useful in reducing the size of the reflectarray antenna by having a short feed distance.

### 3.4. Some Advanced Types of Reflectarray

There are some advanced techniques used in reflectarray antennas that are reported in the literature for highly efficient operation. A work mentioned in [65] proposes a reflectarray of three different types of elements, namely, square patch (SP), ring loaded patch (RLP) and square ring (SR). The best effects of all these elements are combined together to achieve improved gain performance. The ring elements are selected for their wide reflection phase range, and the patch elements have a low loss performance. Collectively, the whole reflectarray, as depicted in Figure 7a, is represented as a high gain unit that is supported by a negligible phase error and a minimum reflection loss. A gain improvement of 1.9 dB was observed using this technique that offers a maximum gain of 29.1 dB. High-efficiency performance can also be obtained along with a dual band operation as proposed in [66], where an FSS (frequency selective surface) is used as an isolation middle layer to separate the two different reflectarrays. This structure acquires maxima of 54% and 50% efficiency with a dual feed operation at X-band and K-band, respectively.

The reflection phase and reflection loss of the reflectarray elements can also be controlled by manually rotating them with a micro-motor [19]. This technique can easily provide efficiency of more than 50% at low frequencies, but with high design complexity. A wideband element with a phase tuning stub or an asymmetric design usually suffers from a high cross polarization that degrades the efficiency of the reflectarray antenna [38,67]. This high cross polarization can be suppressed by mirroring the orientation of the elements on the reflectarray surface. A similar design of a single band reflectarray is shown in Figure 7b, where the elements are mirrored to each other for cross polarization reduction. A three-dimensional (3D) reflectarray element can provide an extra degree of freedom to attain a wide reflection phase range performance with minimum reflection loss. A DRA (dielectric resonator antenna) element and a 3D dipole element are good examples of this technique [68–70]. High design complexity is a major limiting factor in these designs. The metallic losses are removed by using a DRA element, whereas a 3D dipole element offers a good phase matching with the feed for high gain operation. In another strategic type of antenna, the dielectric losses are removed by adopting a full metallic design structure [71,72]. This type of reflectarray is called a fully-metallic reflectarray with a 3D printed non-uniform structure to support the desired progressive phase distribution for high-efficiency operation. However, the cost of the 3D printed dielectric or metallic reflectarray is usually high.

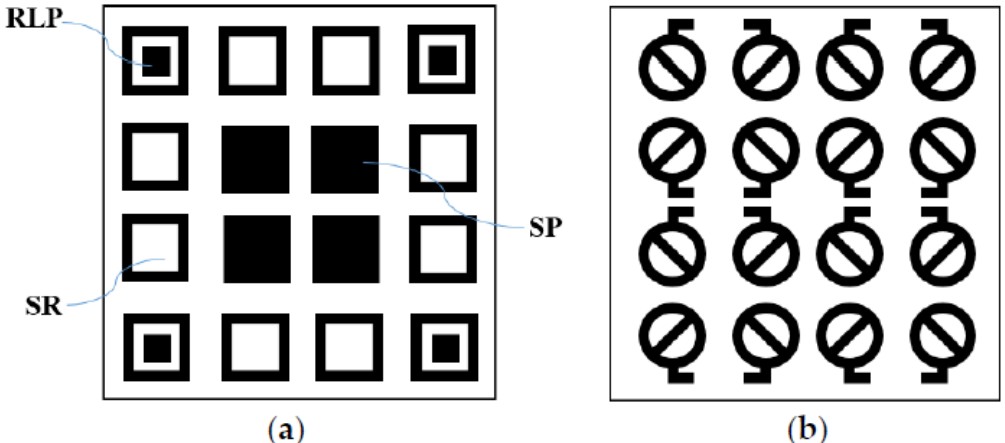

**Figure 7.** High efficiency reflectarray designs: (**a**) a combination of different types elements for gain improvement; (**b**) mirroring of the reflectarray elements for cross polarization reduction.

## 4. Factors Affecting the Aperture Efficiency of the Circular Aperture Reflectarray Antenna

Aperture loss is solely responsible for the prediction of the aperture efficiency of the reflectarray antenna. Similar to the aperture loss, aperture efficiency also possesses a variable nature and can be tuned with respect to the feed distance and feed footprint. The aperture efficiency of a circular aperture reflectarray antenna ($\eta_{apc}$) is the product of its illumination ($\eta_{illc}$) and spillover ($\eta_{sc}$) efficiencies and can be calculated as given in Equation (4).

$$\eta_{apc} = \eta_{illc} \times \eta_{sc} \tag{4}$$

The illumination efficiency defines the aperture area of the reflectarray, which falls within the −3 dB beamwidth of the feed. On the other hand, the concentration of the wasted energy beyond the planar surface of the reflectarray is linked with the spillover efficiency. The quantities of these both efficiencies act alternatively, and a fall in the value of one would give a rise in the other significantly. Both of these quantities can be calculated using the following equations [3].

$$\eta_{illc} = \frac{\left[ \{ (1 - \cos^{q+1} \theta_e)/(q+1) \} + \{ (1 - \cos^q \theta_e)/q \} \right]^2}{2 \tan^2 \theta_e [(1 - \cos^{2q+1} \theta_e)/(2q+1)]} \tag{5}$$

$$\eta_{sc} = 1 - \cos^{2q+1} \theta_e \tag{6}$$

where $\theta_e$ is the half of the subtend angle from the feed to the edge of the reflectarray, as shown in Figure 8, and $q$ is the exponent of the feed pattern function, given as $\cos^q \theta$. This $\theta$ is the −3 dB beamwidth of the feed used in the reflectarray antenna. The equation of the half subtend angle ($\theta_e$) can be derived from the geometry of the reflectarray antenna, as mentioned in Equation (7).

$$\theta_e = \tan^{-1} \left[ \frac{1}{2(f/D)} \right] \tag{7}$$

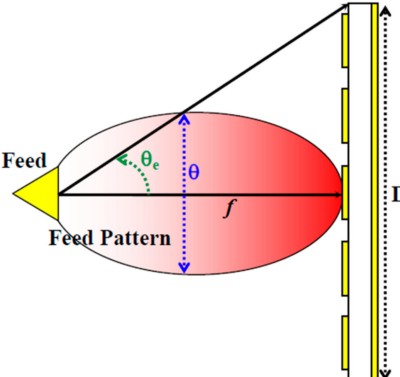

**Figure 8.** Condition for the maximum aperture efficiency of the reflectarray antenna.

Equation (7) shows that the $\theta_e$ depends only on the ratio of the focal length of the feed to the maximum dimension of the reflectarray. In order to get the maximum aperture efficiency from the reflectarray antenna, its half subtend angle should match with the half of the −3 dB beamwidth of the feed ($\theta_e = \theta/2$), as depicted in Figure 8. Assuming that this condition is satisfied, then the value of the feed pattern function ($q$) can be calculated using following relation.

$$q = \frac{-0.15}{\log\left[\cos\frac{\theta}{2}\right]} \tag{8}$$

It has been observed from Equation (8) that the value of the feed pattern function will remain constant even if the feed distance is changed. As indicated in Equations (5) and (6), this feed pattern function can significantly affect the amount of reflectarray aperture efficiency. Equation (8) is derived using an edge tapering of −3 dB in this work, which is actually the half power beamwidth of the feed. An amount of −10 dB of edge tapering can also be used with a modified Equation (8) for a different analysis of the reflectarray aperture efficiency. The same equation can also be modified further for any amount of edge tapering based on the requirements.

### 4.1. Effects of Different Feeds on the Aperture Efficiency

The commonly used feeds in the reflectarray antenna operation are pyramidal horn antennas. Like every antenna type, the gain of the pyramidal horn also depends on its −3 dB beamwidth. Its gain decreases as its beamwidth increases and vice versa. In this analysis, three different pyramidal horns with their respective gains of 10, 15 and 20 dB were selected to study their effects on the aperture efficiency of a circular aperture reflectarray. Their simulated and normalized E-plane radiation patterns at 26 GHz, along with their design specifications, are shown in Figure 9.

Figure 9 shows the three different beamwidths of the pyramidal horn antennas, which are essential to predict the aperture efficiency of the reflectarray. The values of these beamwidths are used to calculate the feed pattern functions using Equation (8), and then Equation (4) is used to predict their overall aperture efficiency with respect to a variable feed distance. The feed distance is taken as a ratio

of the maximum dimension of the reflectarray to the focal length of the feed (*f/D*). Figure 10 contains all the plotted curves for the reflectarray aperture efficiency with three different feeds. It can be seen from Figure 10 that a maximum aperture efficiency of 62% can be achieved with a 10 dB horn as a feed if the *f/D* ratio is set to be at 0.7. This point of maximum aperture efficiency indicates the value of the *f/D* ratio where −3 dB beamwidth (49.6°) of the feed coincide with the half subtend angle. Similarly, for a 15 dB horn as a feed, this value happens to be occurring at an *f/D* ratio of 1.1. In this case, the maximum possible aperture efficiency is 73% with a −3 dB feed beamwidth of 29.3°. That shows that a shorter feed beamwidth requires a longer feed distance to accumulate full aperture of the reflectarray and achieve maximum aperture efficiency. This can also be proved when a 20 dB horn with a −3 dB beamwidth of 14.5° is used as a feed for the reflectarray. The maximum aperture efficiency in this case is 79%, which can be achieved with an *f/D* ratio of 2.1. This analysis shows that the high gain feeds are essential to acquire high reflectarray aperture efficiencies, but with the drawback of a large antenna profile.

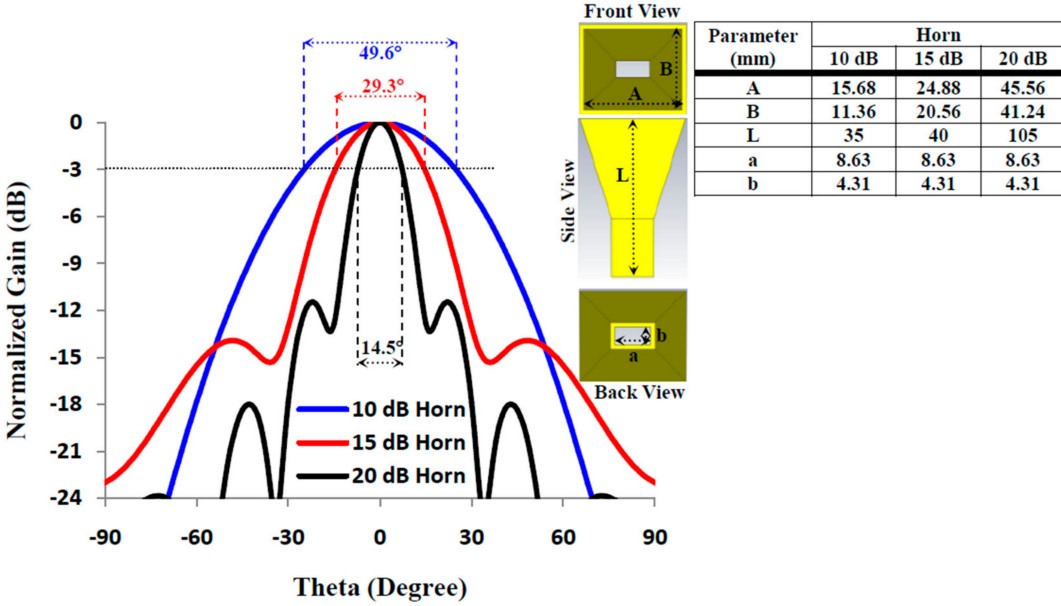

**Figure 9.** E-plane radiation patterns of three different horns along with their dimensions.

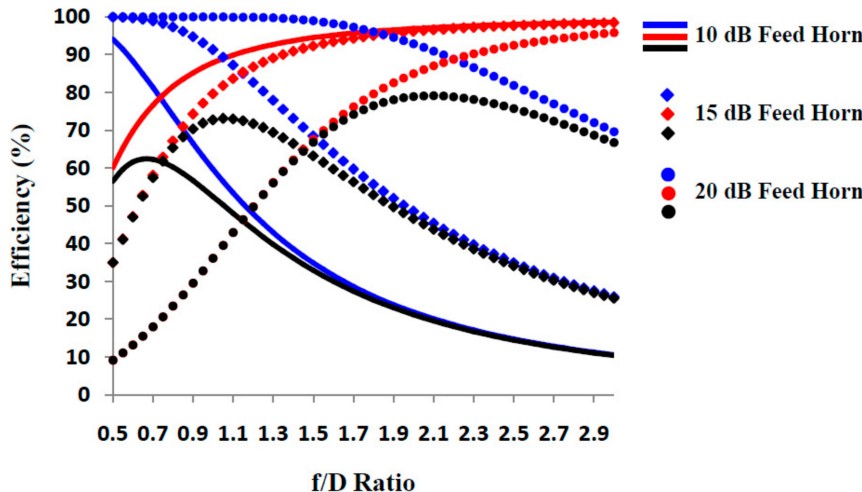

**Figure 10.** Aperture efficiency of the circular aperture reflectarray antenna as a function of its feed distance with three different feeds (colors refer efficiencies as: blue = spillover, red = illumination, black = aperture).

### 4.2. The Effect of Different Feed Distances on the Aperture Efficiency

In another scenario, if the type and beamwidth of the feed are not known for a reflectarray antenna, then the feed can be selected based on a suitable *f/D* ratio with maximum aperture efficiency. Table 2 summarizes the maximum achievable aperture efficiencies with their respective feed beamwidths for different feed distances (*f/D*). It can be observed from Table 2 that each distinct value of the feed distance is linked to a different feed pattern function (q). This feed pattern function is responsible for generating the required beamwidth for the feed in order to get the maximum aperture efficiency. If a compact reflectarray antenna profile is more important than its high aperture efficiency, then wide beamwidth feeds are suitable to be used in it. Alternatively, high aperture efficiency is able to be acquired with a narrow beamwidth feed, but at the cost of a large reflectarray antenna profile. This is the main reason behind the common usage of the feed distances with *f/D* ratio ranging between 0.7 and 1.0. This range of *f/D* ratio provides a good aperture efficiency with a manageable reflectarray antenna profile.

**Table 2.** Selection of the reflectarray feed based on variable feed distance.

| *f/D* | q | Max. Aperture Efficiency (%) | Feed Beamwidth (°) |
|---|---|---|---|
| 0.4 | 2 | 50 | 66 |
| 0.5 | 3 | 57 | 55 |
| 0.6 | 4.5 | 62 | 45 |
| 0.7 | 5.5 | 66 | 41 |
| 0.8 | 7 | 69 | 36 |
| 0.9 | 9 | 71 | 32 |
| 1.0 | 11 | 73 | 29 |
| 1.1 | 13 | 74 | 27 |
| 1.2 | 15 | 75 | 25 |
| 1.3 | 17.5 | 76 | 23 |

### 4.3. The Effect of the Feed Footprint on the Aperture Efficiency

The aperture efficiency analysis provided in Sections 4.1 and 4.2 is based on the center feed reflectarray antenna. However, in case of an offset feed this whole scenario would be different due to the change in the distribution of the feed energy on the reflectarray surface. This phenomenon brings about the need for understanding the feed footprint behavior. Feed footprint is the area illuminated by the feed within its −3dB beamwidth on the surface of the reflectarray. The physical area under the feed footprint depends on the distance (*f/D*) that feed carries from the reflectarray. The center and offset feeds with their plausible footprints are shown in Figure 11. It can be observed from Figure 11 that when the feed is above the center of the reflectarray, its footprint has a circular shape. On the other hand, an offset feed would generate a footprint with an elliptical shape. A reflectarray should cover neither less nor more than the footprint of its feed to attain the maximum aperture efficiency. This scenario shows that a center feed reflectarray will give a maximal aperture efficiency performance if it holds a circular aperture with a suitable feed distance. Alternatively, an elliptical shape reflectarray will be the best option with an offset feed configuration to acquire maximum aperture efficiency.

The factors which affect the feed footprint are the type of the antenna used as the feed and its electric or magnetic field radiation distribution. The feed footprint can differ between an aperture antenna and a wire antenna. All types of horn antennas, dipole antenna, helix antenna and printed antennas, can be used as the feed of the reflectarray. All these types possess different field distributions in their radiation patterns, and hence different feed footprints. Similarly, a dominant E or H field operation of the feed holds a sharp beam in that plane with a narrow beamwidth and spreads the signals in the other plane with a wide beamwidth. Therefore, it is necessary to investigate the field distribution and the footprint behavior of the feed before designing its reflectarray.

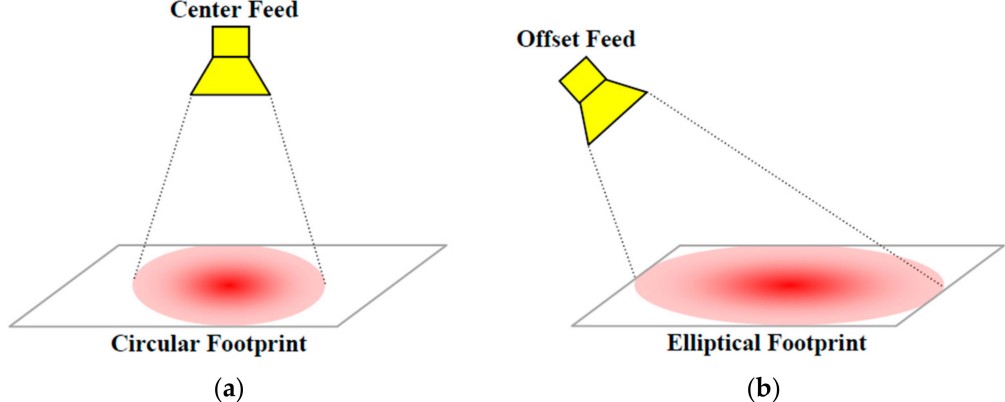

**Figure 11.** Variation in the feed footprint with respect to its position: (**a**) center feed; (**b**) offset feed.

## 5. Aperture Efficiency of the Square Aperture Reflectarray Antenna

The other common aperture shape selected in the reflectarray antenna operation is the square aperture. The behavior of the aperture efficiency in the square aperture reflectarray is totally different than that of the circular aperture reflectarray. The main reason behind it is the change in the physical area of the aperture. This will definitely affect the illumination and spillover efficiencies of the reflectarray antenna in a totally different manner. As shown in Figure 12, a square aperture reflectarray can be set either in a conventional manner or like a diamond shape. If a center feed reflectarray architecture is considered, then these shapes would attain different aperture efficiency performances based on their respective equivalent circular apertures.

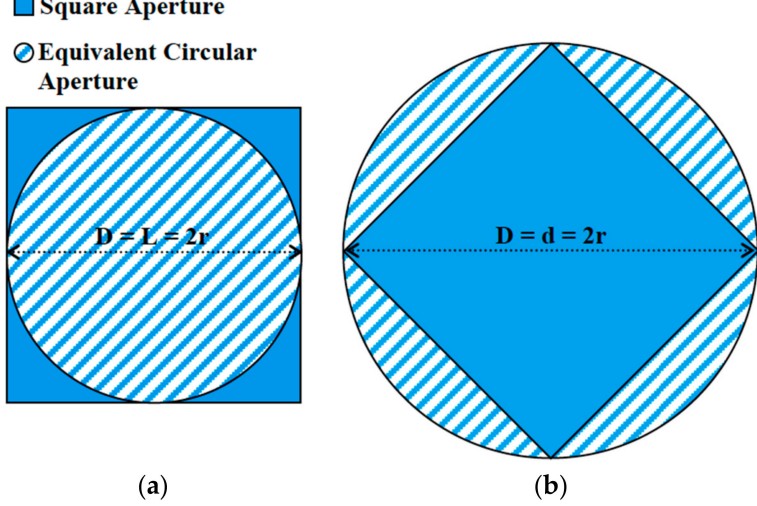

**Figure 12.** Square aperture with its equivalent circular aperture: (**a**) conventional square aperture; (**b**) square aperture in a diamond shape.

### 5.1. Aperture Efficiency of the Conventional Square Aperture

The conventional way to use a square aperture as a reflectarray is shown in Figure 12a. The equivalent circular aperture shown in this case is based on the footprint of a center feed. Both of these apertures actually share the same maximum dimension ($D$), as it equals the length of the square aperture. The respective areas of the square aperture ($A_s$) and its equivalent circular aperture ($A_c$) can be calculated as given below.

$$A_s = L^2 \ and \ A_c = \pi\left(\frac{L}{2}\right)^2 \tag{9}$$

In Figure 12a, the circular aperture shows the area illuminated by the feed over the square aperture. The illuminated area is therefore narrower than the overall physical area of the reflectarray. The percent change between the illuminated and the physical area of the square aperture can be found as derived in Equation (10).

$$\frac{A_c}{A_s} = \frac{\pi\left(\frac{L}{2}\right)^2}{L^2} = \frac{\pi}{4} \tag{10}$$

Equation (10) shows the factor of decrement in the illuminated area of the square aperture and describes the increment in its illumination loss. Therefore, the illumination efficiency of the square aperture reflectarray also decreases by the same factor. On the other hand, its spillover loss remains the same as that of the circular aperture, because no part of the feed footprint is exceeding the boundary of the square aperture reflectarray. Consequently, the illumination ($\eta_{ills}$) and spillover ($\eta_{ss}$) efficiencies of the square aperture reflectarray can be written as in Equation (11).

$$\eta_{ills} = \frac{\pi}{4}\eta_{illc} \ and \ \eta_{ss} = \eta_{sc} \tag{11}$$

The equation of the illumination efficiency of the square aperture reflectarray can also be written as follows:

$$\eta_{ills} = \frac{\pi[\{(1-\cos^{q+1}\theta_e)/(q+1))\} + \{(1-\cos^q\theta_e)/q\}]^2}{8\tan^2\theta_e[(1-\cos^{2q+1}\theta_e)/(2q+1)]}$$

The overall aperture efficiency of the square aperture reflectarray can now be calculated using Equation (12).

$$\eta_{aps} = \eta_{ills} \times \eta_{ss} = \frac{\pi}{4}\eta_{illc} \times \eta_{sc} = \frac{\pi}{4}\eta_{apc} \tag{12}$$

It can be observed from Equation (12) that the aperture efficiency of the reflectarray antenna will decrease by 21.46% if an equivalent square aperture is used instead of the circular aperture.

### 5.2. Aperture Efficiency of the Square Aperture in a Diamond Shape

Another way to consider a square aperture is by tilting it 45° and making a diamond shape, as depicted in Figure 12b. If the incident electric field is horizontally or vertically polarized and aligned with the unit cell element, then the maximum dimension ($D$) of the square aperture equals its diagonal ($d$). The feed is supposed to illuminate all the square aperture along its maximum dimension by generating a circular footprint. The area under this circular footprint ($A_c$) is larger than the physical area of the square aperture ($A_s$) and can be estimated as described below.

$$A_s = \frac{d^2}{2} \ and \ A_c = \pi\left(\frac{d}{2}\right)^2 \tag{13}$$

It can be observed from Figure 12b that the area under the square aperture is less than that of the equivalent circular aperture. Equation (14) describes the percent change in area between the square aperture and its equivalent circular aperture.

$$\frac{A_s}{A_c} = \frac{d^2/2}{\pi\,(d/2)^2} = \frac{2}{\pi} \tag{14}$$

In this case, the accession of the illuminated footprint will generate extra spillover losses due to the loss of the incident signals beyond the boundaries of the square aperture. This phenomenon will drastically decrease the spillover efficiency of the square aperture reflectarray. Alternatively, its illumination efficiency will remain the same because all of the physical area of the square aperture is

illuminated by the feed. Therefore, in this case, the illumination ($\eta_{ills}$) and spillover ($\eta_{ss}$) efficiencies of the square aperture reflectarray can be calculated as given in the following expression.

$$\eta_{ills} = \eta_{illc} \ and \ \eta_{ss} = \frac{2}{\pi}\eta_{sc} \tag{15}$$

The spillover efficiency of a square aperture reflectarray in a diamond-like shape can also be expressed as given below.

$$\eta_{sc} = \frac{2\left(1 - \cos^{2q+1}\theta_e\right)}{\pi}$$

Equation (16) describes the overall aperture efficiency of a square aperture reflectarray in a diamond-like shape.

$$\eta_{aps} = \eta_{ills} \times \eta_{ss} = \eta_{illc} \times \frac{2}{\pi}\eta_{sc} = \frac{2}{\pi}\eta_{apc} \tag{16}$$

Equation (16) shows that the aperture efficiency of a diamond-like square aperture reflectarray antenna is 36.33% less than that of an equivalent circular aperture reflectarray antenna.

### 5.3. Aperture Efficiency Comparison between Circular and Square Aperture Reflectarrays

The mathematical expressions derived in Equations (12) and (16) show that the aperture efficiency of the reflectarray antenna depends on the selection of its aperture shape. Figure 13 compares the predicted aperture efficiency of a circular aperture reflectarray antenna with its equivalent square aperture reflectarray antenna. The equivalent aperture is taken in terms of its maximum dimension (*D*). It can be seen from Figure 13 that for each different feed horn the maximum aperture efficiency occurred at its respective feed distance (*f/D*), regardless of its different aperture shape. The feed distance related to the maximum aperture efficiency differs only with different feed types. This is because of the different beamwidths of each feed used in the aperture efficiency analysis, as already discussed in Section 4.1, although the amount of the maximum aperture efficiency at a certain feed distance changes with the change in shape of the aperture.

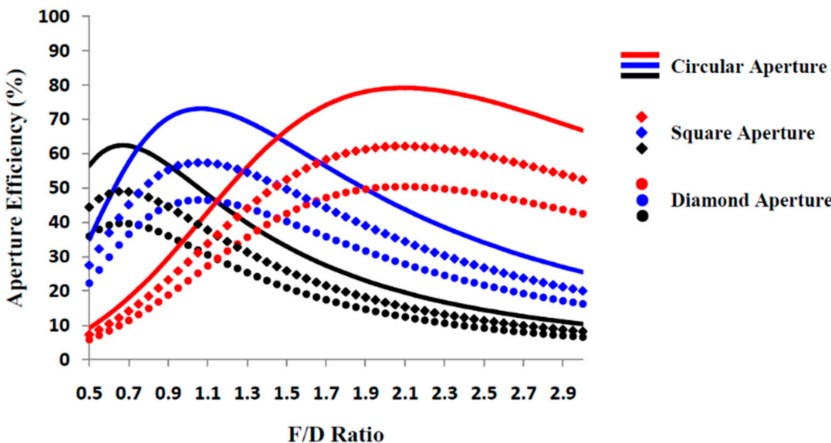

**Figure 13.** Comparison of the aperture efficiency for different reflectarray apertures with different feeds having variable feed distance (colors refer feeds as: black = 10 dB, blue = 15 dB, red = 20 dB).

The circular aperture reflectarray with a 10 dB feed horn attains a maximum aperture efficiency of 62%, but it decreases to 49% and 39% when equivalent square and diamond aperture reflectarrays are used, respectively. The same trend can also be noticed when a 15 dB or a 20 dB feed horn is used in the reflectarray operation at its respective feed distance. The reduction in the aperture efficiency of the equivalent square aperture reflectarray is due to the reduction in its illumination efficiency, while its spillover efficiency remains the same that as of the circular aperture reflectarray. The spillover

efficiency of the equivalent diamond aperture is less than that of the circular aperture, whereas its illumination efficiency is unchanged as compared to the circular aperture reflectarray. That means, in order to get a higher efficiency performance, the maximum dimension (*D*) of a square aperture reflectarray should be taken as its length of the side instead of its diagonal length.

Other than these aperture shapes, a reflectarray could also possesses a hexagonal or octagonal shape, which in reality both behave like a circular aperture. That is why the hexagonal and octagonal shapes are not very common in reflectarray antenna design. Additionally, just like a square aperture, a reflectarray could also have a rectangular aperture, as discussed in [73]. The rectangular reflectarray aperture is often used as a sector antenna to produce a fan beam that provides ease of beamsteering in one scanning plane. However, the rectangular aperture has a very poor aperture efficiency performance because it offers very high spillover loss. As in the case of [73], the rectangular aperture reflectarray offers a very poor efficiency performance of just 3.3%. Therefore, only three examples of aperture shapes, namely, circular, square and diamond, are discussed in this work because these aperture shapes provide the best reflectarray performance in terms of the aperture efficiency. Moreover, all the other aperture shapes of the reflectarray are actually extracted from or related to these three types of the aperture shapes.

## 6. A Practical Method to Accurately Predict and Measure the Efficiency of a Reflectarray Antenna

The aperture efficiency analysis of the reflectarray antenna provided in the previous section is purely based on the behavior of its illumination loss and spillover loss. The prediction of the reflectarray aperture efficiency is possible even without the consideration of the type of its unit cell patch element and resonance behavior. However, as discussed in Section 2, there are some other sources of the loss in reflectarray antenna related to its resonance response, which can drastically affect its efficiency performance. Therefore, the total efficiency of the reflectarray antenna is always lower than its aperture efficiency. The total efficiency ($\eta_t$) of the reflectarray antenna can be divided into the feed efficiency ($\eta_f$), patch efficiency ($\eta_p$) and aperture efficiency ($\eta_{ap}$), as given in Equation (17).

$$\eta_t = \eta_f \times \eta_p \times \eta_{ap} \tag{17}$$

Each type of the efficiency mentioned in (17) is associated with the portion of its loss performance in the reflectarray antenna. Consequently, it is necessary to properly analyze the performance of the feed and patch element before designing its reflectarray.

### 6.1. Design and Analysis of the Pyramidal Horn Feed

The most common feed used in the reflectarray design is the pyramidal horn antenna. The pyramidal horn feed used in this work is based on the principle of a standard horn antenna that can be used within the frequency band of 22–33 GHz. The fabricated pyramidal horn is shown in Figure 14a, and its dimensions were already mentioned in Figure 9 as the 10 dB feed horn. As depicted in Figure 14b, its loss performance has been measured with the help of a WR-34 coaxial to waveguide adapter in the frequency range between 24 and 28 GHz. The return loss shown in Figure 14b describes the resonance behavior of the pyramidal horn. It can be observable from Figure 14b that the return loss is considerably lower than the standard resonant benchmark of −10 dB level. It confirms a smooth resonating performance of the feed horn within the selected frequency band of 24–28 GHz. On the other hand, the feed loss is associated with the loss of energy inside the pyramidal horn during its operation. Good resonance behavior with negligible losses can be observed for the pyramidal horn within the selected frequency range. Due to very low loss performance of the pyramidal horn, feed losses in the reflectarray operation are often neglected.

The radiation characteristics of the pyramidal horn were also measured and compared with its ideal performance, which can be seen in Figure 15. It is obvious from Figure 15 that the fabricated pyramidal horn attains a shorter −3 dB beamwidth with a higher gain performance as compared to its

predicted response. This discrepancy in the measured results was mainly due to the high sensitivity of the dimensions selected for the fabrication in the required frequency range. The effect of this discrepancy can be avoided by selecting the measured characteristics of the pyramidal horn for the reflectarray design.

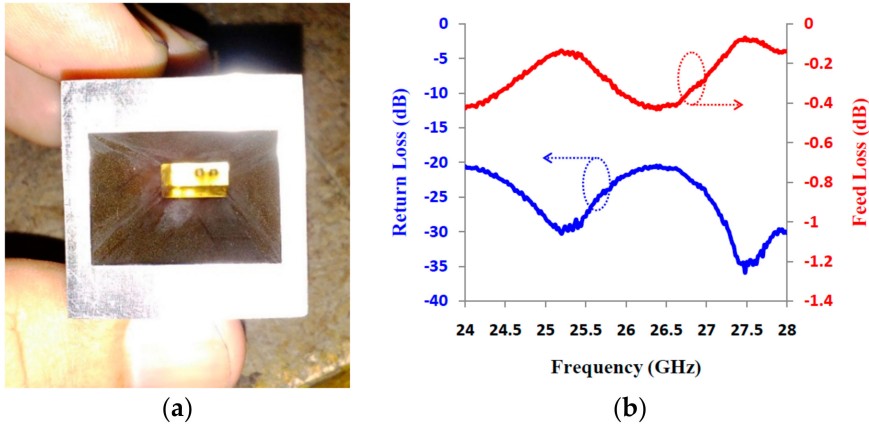

(**a**)                                                    (**b**)

**Figure 14.** (**a**) Fabricated pyramidal horn antenna; (**b**) measured loss performance of the pyramidal horn antenna.

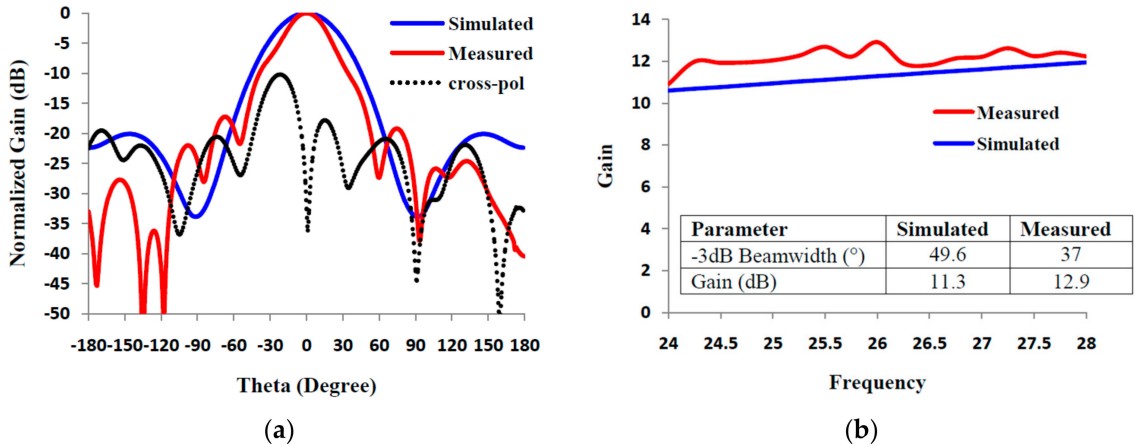

(**a**)                                                    (**b**)

**Figure 15.** Simulated and measured radiation characteristics of the pyramidal horn antenna: (**a**) E-plane radiation pattern with cross polarization measured at 26 GHz; (**b**) gain versus frequency.

*6.2. Design Characteristics of the Unit Cell Patch Element*

The square patch unit cell element was selected because of its simple design and ease of fabrication at the high frequency of 26 GHz. The patches were printed above a 0.25 mm thick Rogers RT/d 5880 substrate with a dielectric constant of 2.2. In order to avoid the mutual coupling between the adjacent patch elements, an element spacing of $\lambda/2$ (5.77 mm) was taken into account. A waveguide simulator was designed to coincide with dimensions of the two-patch-unit cell element (11.54 mm × 5.77 mm) with the aperture of a WR-34 adapter (8.63 × 4.31 mm). The overall measurement setup is shown in Figure 16a, where the open end of the waveguide is used to hold the two-patch-unit cell element. The waveguide simulator is designed to perform the scattering parameter measurements of the reflectarray unit cell with infinite boundary conditions [74,75]. Figure 16b shows the fabricated two-patch-unit cell elements with a variable patch length indicated above each sample.

The simulated and measured results of unit cell elements are compared in Figure 17. It can be observed that the simulated and measured results follow the same trend with a progressive phase range of 330°, which holds a phase error of 30°. Phase error directly affects the gain and efficiency performance of the reflectarray antenna. Its effect can be minimized if the reflectarray antenna is designed with

precision. On the other hand, despite offering a same resonance performance, the measured loss was about 1 dB higher than the simulated loss. This additional loss is related to the cable, connectors and the propagation effects inside the waveguide simulator. This extra loss cannot be avoided during the unit cell measurements. However, it can be neglected in the analysis of the reflectarray antenna where the cable, connectors and waveguide are not used.

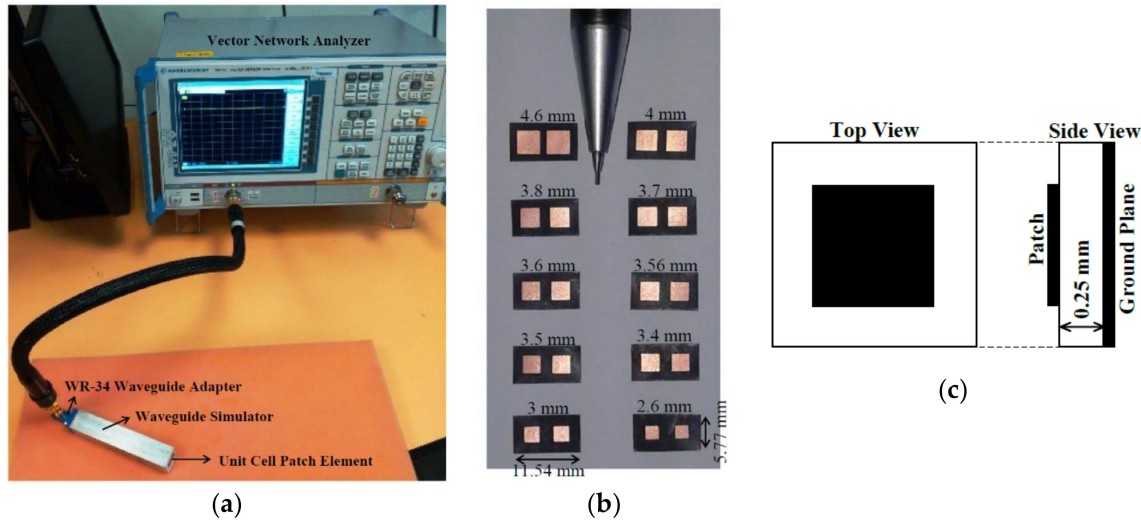

**Figure 16.** (**a**) Measurement setup for scattering parameter measurements; (**b**) fabricated square patch unit cell elements; (**c**) top and side view of the square patch element.

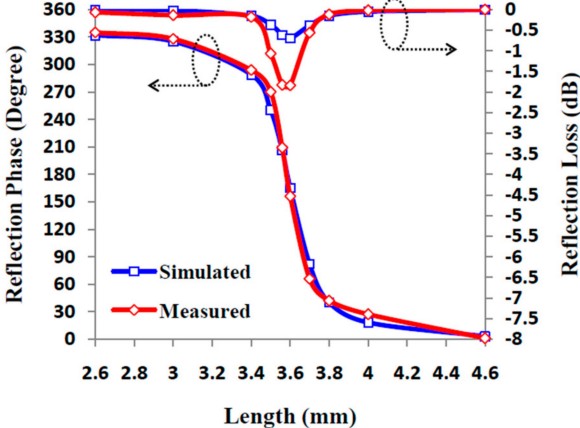

**Figure 17.** Reflection response of the square patch unit cell element with variable length at 26 GHz.

*6.3. Design and Validation of the Circular and Square Aperture Reflectarrays*

The circular and square aperture reflectarrays have been designed and fabricated with the equivalent apertures in terms of their respective maximum aperture dimensions (*D*). Figure 18a depicts the circular and square aperture reflectarrays with the graphical representation of the progressive phase distribution related to the quadrants of the respective reflectarrays. The circular aperture reflectarray consists of the 332 variable size square patch elements, whereas the square aperture holds the 20 × 20 array of the same patch elements. The diameter of the circular aperture and the length of the square aperture reflectarrays have been taken as 10*λ* (115.4 mm). As shown in Figure 18b, a dielectric frame has been designed to properly hold the feed and the reflectarray during the far-field measurements. As has already been shown in Figure 15b, the selected pyramidal horn feed has a −3dB beamwidth of 37°. Therefore, in order to fully accumulate the reflectarray aperture with minimal aperture losses, the feed has been set at an *f/D* ratio of 0.8, as mentioned in Table 2.

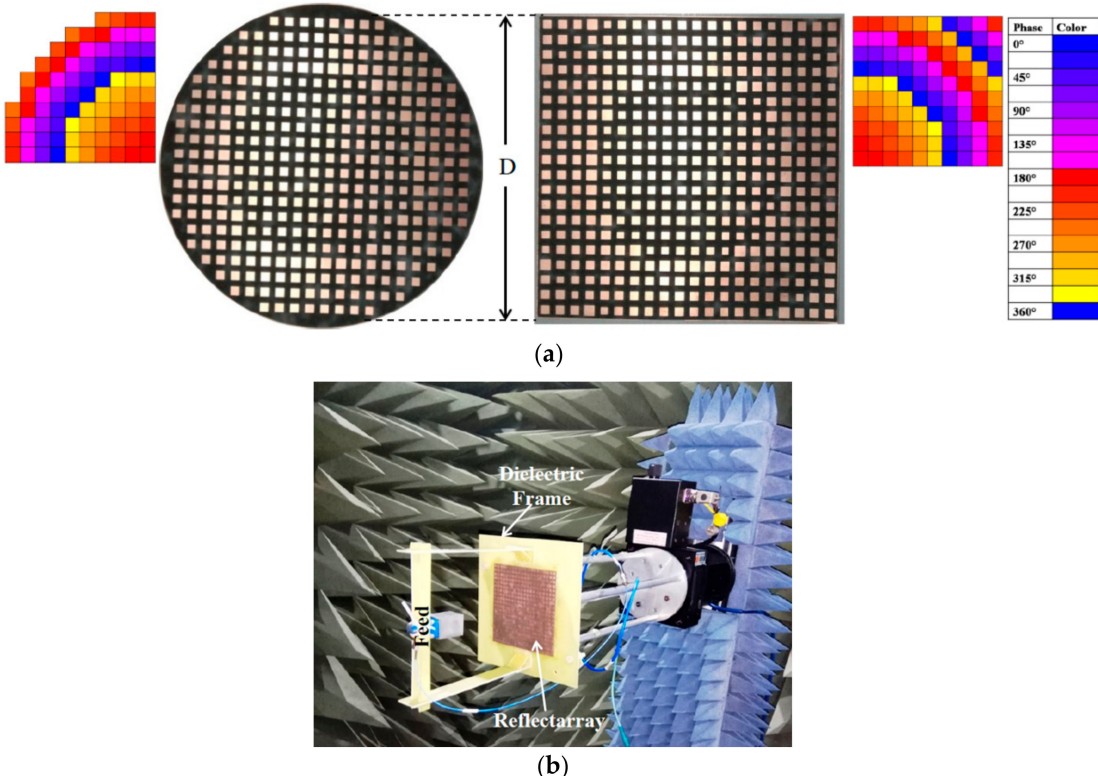

(a)

(b)

**Figure 18.** (**a**) Circular and square aperture reflectarrays with their progressive phase distribution; (**b**) reflectarray during measurements.

The comparison between the simulated and measured radiation patterns of the selected reflectarrays has been plotted in Figure 19. It can be observed that the simulated and measured radiation pattern curves for both selected reflectarrays follow the same trend despite a minor discrepancy in the side lobe levels. However, the side lobe levels of the both selected reflectarrays are still acceptable because they lay well below the maximum allowable level of −10 dB. A low side lobe level authenticates the proper designing of reflectarray antenna with negligible effect of phase error on the collimation of reflected signals. The measured response of the cross polarization for both reflectarrays is also less than −25 dB in the broadside direction. The gain of the selected reflectarrays was also measured and compared with the simulated results, as depicted in Figure 20. It can be seen that the measured and simulated gain versus frequency responses are well matched, especially at the resonant frequency (26 GHz). Further discussions on these results are provided in the next section.

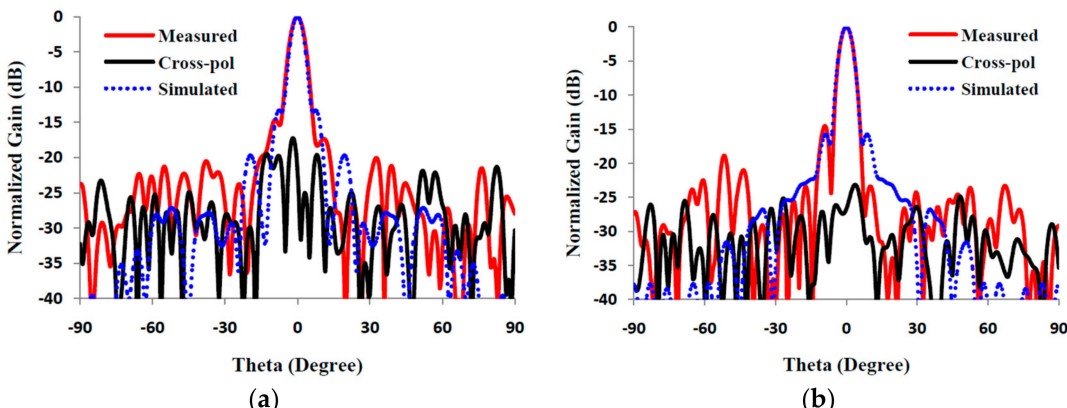

(a)                                              (b)

**Figure 19.** Measured and simulated E-plane radiation patterns with measured cross polarization of the selected reflectarrays at 26 GHz: (**a**) circular aperture reflectarray; (**b**) square aperture reflectarray.

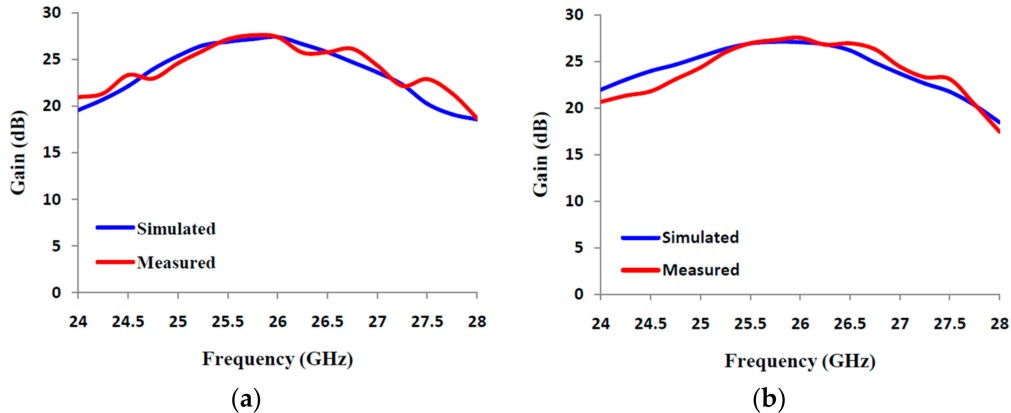

**Figure 20.** Measured and simulated gain versus frequency performances of the selected reflectarrays: (**a**) circular aperture reflectarray; (**b**) square aperture reflectarray.

*6.4. Efficiency Prediction by Gain-Directivity Relation*

The dependency of the antenna efficiency on its gain and maximum directivity is discussed in detail in Section 2. The maximum directivity of the reflectarray antenna can be calculated using the formula given in Equation (2). In order to estimate the efficiency of the reflectarray antenna, its gain value is required either in simulated or in measured [76] form. The simulated and measured values of the gain of the selected reflectarray apertures with their respective efficiencies at 26 GHz are summarized in Table 3. The efficiencies mentioned here are referred to as the total efficiency of the reflectarray antenna. The efficiency is calculated either from the simulated or from the measured gain value, using the relation given in Equation (3). It has been noticed from Table 3 that the measured efficiencies for reflectarray apertures were higher than those of their simulated counterparts. This is because of a slightly higher measured gain as compared to the simulated gain for both reflectarray apertures. The reason behind the slightly higher measured gain is the shorter measured −3 dB beamwidth of the feed as compared to its simulated beamwidth, as already mentioned in Figure 15b. This shows that the feed characteristics in the simulations and measurements of the selected reflectarray apertures are slightly different. Moreover, the efficiency of the square aperture reflectarray can also be predicted using Equation (12), if the efficiency of its equivalent circular aperture reflectarray is known. Consequently, the efficiencies of the square aperture reflectarray can be calculated as 40.2% and 42.7% from the simulated and measured efficiencies of the circular aperture reflectarray, respectively. These two predicted values of the square aperture reflectarray efficiencies are almost the same as its actual simulated and measured efficiencies mentioned in Table 3. Furthermore, if the same square aperture reflectarray is set to operate in a diamond-like shape, then using Equation (16) its efficiency will further decrease to 32.6% and 34.6% for simulated and measured efficiencies of the circular aperture reflectarray respectively. That is why it is not recommended to use a square aperture reflectarray in a diamond-like shape with its diagonal considered as its maximum reflectarray dimension.

**Table 3.** Measured and simulated efficiencies of the selected reflectarray apertures at 26 GHz.

|  | Parameter | Circular Aperture | Square Aperture |
| --- | --- | --- | --- |
| Predicted | Max. Directivity (dB) | 29.94 | 31 |
| Simulated | Gain (dB) | 27.03 | 27.2 |
|  | Efficiency (dB) | −2.91 (51.2%) | −3.8 (41.7%) |
| Measured | Gain (dB) | 27.3 | 27.4 |
|  | Efficiency (dB) | −2.64 (54.4%) | −3.6 (43.6%) |

### 6.5. Efficiency Prediction by Loss Quantification

Another way to estimate the reflectarray antenna efficiency by its loss quantification using Equation (17) is presented here. This method shows how the efficiency of the reflectarray antenna is affected by the various forms of its loss performance. All the possible sources of reflectarray losses with their simulated and measured values are listed in Table 4. It should be noted that the cross polarization loss is neglected here due to its insignificant effect, as already measured in Figure 19. The feed and patch losses are taken from their simulated and measured results. As discussed in Section 6.2, the waveguide and connector losses are not considered here in the measured value of the patch loss. The feed loss and patch loss are the same for both reflectarray apertures, whereas aperture loss differs with the shape of the aperture. The illumination and spillover losses represent the aperture loss of the respective reflectarray aperture. The simulated and measured values of the illumination and spillover losses were calculated using the simulated and measured response of the feed horn respectively. Additionally, the illumination and spillover losses of the circular aperture reflectarray antenna are calculated using Equations (5) and (6) respectively. On the other hand, Equation (11) was used for the calculation of the same losses for the square aperture reflectarray antenna. The total loss estimated in this method should match with the gain-directivity difference identified in Table 3. The simulated and measured total losses estimated here for the circular aperture reflectarray antenna are −3.01 and −2.8, which are comparable with their respective values in Table 3. The same scenario can also be observed in the case of the square aperture reflectarray antenna. Moreover, the predicted efficiencies of the square aperture reflectarray using Equation (12) from the simulated and measured efficiencies of the circular aperture reflectarray are 39.3% and 41.2%, respectively. These estimated values are almost same as the simulated and measured efficiencies of the square aperture reflectarray mentioned in Table 4. That proves the authenticity of the mathematical relation derived in Equation (12).

**Table 4.** Quantification of the loss performance for the selected reflectarray apertures at 26 GHz.

| Type of Loss | Circular Aperture | | Square Aperture | |
|---|---|---|---|---|
| | Simulated | Measured | Simulated | Measured |
| Feed (dB) | −0.1 | −0.36 | −0.1 | −0.36 |
| Patch (dB) | −0.7 | −0.84 | −0.7 | −0.84 |
| Illumination (dB) | −0.91 | −1.19 | −1.96 | −2.21 |
| Spillover (dB) | −1.3 | −0.41 | −1.3 | −0.41 |
| Total (dB) | −3.01 | −2.8 | −4.06 | −3.82 |
| Total Efficiency | 50% | 52.5% | 39.3% | 41.5% |

## 7. Conclusions

The efficiency of the reflectarray antenna depends on the type and amount of losses it contains. It can be divided into feed, patch and aperture efficiencies. The feed and patch efficiencies are related to the frequency of operation and are independent of the size and shape of the reflectarray aperture. Consequently, the aperture efficiency can be enhanced by optimizing the reflectarray feeding mechanism, which is related to the type and radiation characteristics of the feed. Additionally, the shape of the reflectarray aperture is also an important factor to be considered in the proper efficiency analysis. The shape and size of the reflectarray should coincide with the feed footprint to attain a maximum aperture efficiency. The reflectarray aperture efficiency can be estimated using mathematical equations associated with the size and the shape of the reflectarray aperture. The illumination and spillover losses, which are related to the aperture efficiency, are governed by the shape of the reflectarray aperture. A conventional square aperture holds 21.46% higher illumination loss as compared to its equivalent circular aperture reflectarray. On the other hand, spillover loss increases by 36.33% when the same square aperture reflectarray is held like a diamond shape. A higher aperture loss results in a lower aperture efficiency, which tends to reduce the total efficiency of the reflectarray antenna. All the losses associated with the reflectarray antenna can be simulated and measured to estimate the total

efficiency of reflectarray antenna. Moreover, the total efficiency of the square aperture can be estimated using the derived mathematical equations if the efficiency of its equivalent circular aperture is known. The concept of estimating and improving the efficiency, provided in this paper, can be easily applied on any shape and type of the reflectarray antenna.

**Author Contributions:** Conceptualization, M.H.D. and M.H.J.; methodology, M.H.D. and M.I.A.; software, M.H.D., M.I.A. and A.Y.I.A.; validation, M.H.J., F.C.S. and M.R.K.; formal analysis, M.H.D. and M.I.A.; investigation, M.H.D. and N.F.S.; resources, M.H.J. and F.C.S.; data curation, M.H.D., A.Y.I.A. and N.F.S.; writing—original draft preparation, M.H.D. and M.I.A.; writing—review and editing, M.H.J., F.C.S. and M.R.K.; visualization, M.H.J. and M.R.K.; supervision, M.H.J. and F.C.S.; project administration, M.H.J. and M.R.K.; funding acquisition, M.H.J. and M.R.K. All authors have read and agreed to the published version of the manuscript.

**Funding:** This work is supported in part by the Ministry of Education Malaysia, Ministry of Science Technology and Innovation, the Research Management Center Universiti Tun Hussein Onn Malaysia under research fund E15501, and Universiti Teknologi Malaysia under research grants Vot 4J415, Vot 03G33, Vot 4S134 and Vot13H26.

**Acknowledgments:** The authors wish to thank the staff of the Wireless Communication Centre, Universiti Teknologi Malaysia, for the technical support.

**Conflicts of Interest:** The authors declare no conflict of interest.

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
