# Peer review of "Aspects of Efficiency Enhancement in Reflectarrays with Analytical Investigation and Accurate Measurement"

_electronics, doi:10.3390/electronics9111887_

Round 1
Reviewer 1 Report
The authors have made the necessary modification for providing a more complete overview of the subject.
Reviewer 2 Report
The paper illustrates an overview of the different solutions adopted in the literature to increase the reflectarray efficiency. Furthermore, the authors perform a losses quantification of reflectarray antenna and discuss the factors affecting the aperture efficiency of reflectarray antenna.
The paper is interesting, however, in my opinion there is not sufficient advancement with respect to previous published works. As a matter of the fact, a comprehensive analysis of reflectarray aperture efficiency and losses has already been illustrated in other previous published works, such as [34] and [3]:
[34] Pozar, D.M.; Targoski, D.; Syrigos, H.. D.; Targonski, S.D.; Syrigos, H.. D. Design of millimeter 819 wave microstrip reflectarrays. IEEE Trans. Antennas Propag. 1997, 45, 287–296, 820 doi:10.1109/8.560348;
[3] Huang, J.; Encinar, J. Reflectarray antennas; Wiley Inter Science: USA, 2007.
The only appreciable novelty with respect to reference [34] lies in the evaluation of the Aperture Efficiency of the Square Aperture Reflectarray Antenna (see Section 5).
Furthermore, the document is redundant and hard to read, it looks more like a thesis or a technical report than a scientific work.
Taking into account the above criticisms, I would recommend to revise the manuscript and to submit a properly reduced version.
Reviewer 3 Report
The paper is basically containing the theory for efficiency improvement with some basic calculations being done for the predictions of simulated or measured efficiencies. In theory, different perspectives of losses, i.e., patch losses, aperture losses, and feed losses, and their effect on performance of reflectarray are studied thoroughly. Finally, the authors have compared the performance of reflectarray with circular and square aperture that are constructed with square patch unit cells through simulations, measurements, and investigated analytically. Although nothing new in the paper, but the originality of the work lies in the detail and thorough explanation of the reflectarray performance. Therefore, I recommend acceptance of this paper for publicaton.Author Response
Please see attachment

Round 2
Reviewer 2 Report
Taking into account that in the first round of revision the authors was suggested to convert their manuscript into a review paper, by providing more information and including some other related topics, I agree to publish the paper as it is.
However, I would recommend the authors to rewrite the abstract, in order to clarify and highlight the scope of their paper, namely that it is a review paper on new techniques and projects for reflectarray efficiency improvement.
Whilst, their personal and innovative contribution must be presented only at the end of the abstract (i.e. Equation for the evaluation of aperture efficiency of a square aperture reflectarray; practical method to accurately predict and measure the efficiency of a reflectarray antenna, etc.)
Author Response
Please see the attachment
This manuscript is a resubmission of an earlier submission. The following is a list of the peer review reports and author responses from that submission.
Round 1
Reviewer 1 Report
This paper studies the loss and the efficiency of the reflectarray antennas. It is generally well written, while the followings need to be considered in the revision.
- The upper part of Abstract is a little bit verbose. Please paraphrase.
- Figures are of high quality, which makes this paper attractive.
- For better readability, it would be better to magnify the table in Figure 6.
- More of the latest works should be introduced. References are a bit old-fashioned.
Reviewer 2 Report
The authors provided a very detailed analysis on the aspects that affect the efficiency in reflectarrays. However, in my opinion, these analysis are trivial and the results don't provide much additional value in antenna design. The detailed comments are listed below.
- The authors didn't clearly clarify the novelty in this work.
- In section 5, in antenna design example discussions. The aspects that affect the antenna performance are all well known. The discussions are not related to the analysis in the previous sections.
- The results shown in section 5 are nothing more than a comparison between simulation and measurement. I don't see how the analysis in the previous section affect the design of the antenna.
- For a thorough analysis, only three examples are not sufficient. If the authors want to prove the correctness of the analysis, a statistics results with a larger number number of examples are necessary.
Reviewer 3 Report
The manuscript presents a study on the efficiency of reflectarrays, primarily focused on geometrical (i.e. aperture based) considerations. The presentation is clear and sound. Main question that rises is why the authors insist on differentiating between "square" and "diamond" as if they were different geometrical shapes whereas in reality they consider different illumination conditions.Author Response
Please see the attachment

Reviewer 4 Report
This is an interesting analysis about reflectarrays that, while does not cover any aspect of reflectarrays, can be of interest as an introductory reading, as well as for approaching didactic of reflectarrays
Reviewer 5 Report
A general discussion on the efficiency of reflectarray antennas is presented. The work is well written, but it neglected a considerable number of previous research works on the same topic. The paper cannot be considered an original research work but may be a review manuscript once that the the following references are opportunely included in the bibliography and discussed. Indeed, ref. 1 reports formulas for evaluating the efficiency of reflectarray antennas. Ref.2-6 report relevant results about loss reduction in reflectaray panels by using subwavelength elements. In particular, it has been demonstrated that, using subwavelength elements allows to drastically reduce reflection losses. These aspects are not discussed in the presented paper and therefore it should be published in this form.
- Yu, A., Yang, F., Elsherbeni, A. Z., Huang, J., & Rahmat‐Samii, Y. (2010). Aperture efficiency analysis of reflectarray antennas. Microwave and Optical Technology Letters, 52(2), 364-372.
- Pozar, D. M. (2007). Wideband reflectarrays using artificial impedance surfaces. Electronics letters, 43(3), 148-149.
- Ethier, J., Chaharmir, M. R., Shaker, J., & Lee, D. (2012). Development of novel low-cost reflectarrays [antenna applications corner]. IEEE Antennas and Propagation Magazine, 54(3), 277-287.
- Costa, F., & Monorchio, A. (2012). Closed-form analysis of reflection losses in microstrip reflectarray antennas. IEEE transactions on antennas and propagation, 60(10), 4650-4660.
- Ethier, J., Chaharmir, M. R., & Shaker, J. (2012). Loss reduction in reflectarray designs using sub-wavelength coupled-resonant elements. IEEE transactions on antennas and propagation, 60(11), 5456-5459.
- Ethier, J., Chaharmir, M. R., & Shaker, J. (2012). Reflectarray thinning using sub-wavelength coupled-resonant elements. Electronics letters, 48(7), 359-360.
Round 2
Reviewer 2 Report
I still don't see the significance of this work in the revised manuscript. In additional to the added claims of the novelty. I don't see how this work can "help researchers to significantly save their design time" for "any type of reflectarray antenna" through out the manuscript.
Reviewer 5 Report
In the reviewed version of the manuscript, the authors included the following sentence while discussing about subwavelength elements reflectarray:
“However, the sub-wavelength elements can produce a large amount of mutual coupling between the elements if they are not designed with precision [3]. The mutual coupling is produced due to reduction in element spacing with increment in capacitive loading between the elements. This mutual coupling can generate grating lobes in the far-field response of the reflectarray antenna, which is not a good sign for a high gain reflectarray performance.”
- The mutual coupling among the elements in not an issue as the elements are designed by using Floquet approach which already take into account the effect of coupling.
- The mutual coupling does not generate grating lobes and it correspond to a tight sampling of the surface. On the contrary, largely spaced elements can generate grating lobes.